# Overview of Prostaglandin E2 (PGE2)-Targeting Radiolabelled Imaging Probes from Preclinical Perspective: Lessons Learned and Road Ahead

**DOI:** 10.3390/ijms24086942

**Published:** 2023-04-08

**Authors:** Zita Képes, Noémi Dénes, István Kertész, István Hajdu, György Trencsényi

**Affiliations:** Division of Nuclear Medicine and Translational Imaging, Department of Medical Imaging, Faculty of Medicine, University of Debrecen, Nagyerdei St. 98, H-4032 Debrecen, Hungary

**Keywords:** Bismuth-205/206 (^205/206^Bi), cyclodextrins (CDs), Gallium-68 (^68^Ga), 2-hydroxypropyl-β-cyclodextrin (HPBCD), positron emission tomography (PET), prostaglandin E2 (PGE2), randomly methylated β-CD (RAMEB)

## Abstract

As malignancies still represent one of the major health concerns worldwide, early tumor identification is among the priorities of today’s science. Given the strong association between cyclooxygenase-2 (COX-2)/prostaglandin E2 (PGE2), PGE2 receptors (EPs), and carcinogenesis, target-specific molecules directed towards the components of the COX2/PGE2/EP axis seem to be promising imaging probes in the diagnostics of PGE2pos. neoplasms and in the design of anti-cancer drugs. Featured with outstanding inclusion forming capability, β-cyclodextrins (CDs) including randomly methylated β-CD (RAMEB) were reported to complex with PGE2. Therefore, radiolabelled β-CDs could be valuable vectors in the molecular imaging of PGE2-related tumorigenesis. In vivo preclinical small animal model systems applying positron emission tomography (PET) ensure a well-suited scenario for the assessment of PGE2-affine labelled CD derivatives. Previous translational studies dealt with the evaluation of the tumor-homing capability of Gallium-68 (^68^Ga) and Bismuth-205/206 (^205/206^Bi)-appended β-CD compounds conjugated with chelator NODAGA or DOTAGA: [^68^Ga]Ga-NODAGA-2-hydroxypropyl-β-cyclodextrin/HPBCD, [^68^Ga]Ga-NODAGA-RAMEB, [^68^Ga]Ga-DOTAGA-RAMEB, and [^205/206^Bi]Bi-DOTAGA-RAMEB in experimental tumors with different PGE2 expression. These imaging probes project the establishment of tailor-made PET diagnostics of PGE2pos. malignancies. In the present review, we provide a detailed overview of the in vivo investigations of radiolabelled PGE2-directed CDs, highlighting the importance of the integration of translational discoveries into routine clinical usage.

## 1. Introduction

### 1.1. Cyclooxigenase-2 (COX2)/Prostaglandin E2 (PGE2)/PGE2-Receptor (EP) Axis

Given the strong association between cyclooxygenase-2 (COX-2)/prostaglandin E2 (PGE2), PGE2 receptors (EPs), and carcinogenesis, an immense number of research studies has been spawned to intensively investigate the comprehensive molecular mechanisms behind [1]. 

Membrane-attached, inducible COX-2 is the key rate-limiting enzyme in the conversion of arachidonic acid into prostaglandins [1]. Literature data indicate that COX-2 overexpression seems critical in cancer initiation, tumor progression, and tumor maintenance [2]. Further, prior studies reported COX-2 upregulation in different cancer types, such as glioblastoma, colon, breast, prostate, and urinary bladder cancer [3,4,5,6,7]. Hence, COX2 specific inhibitors (COXibs) or non-steroidal anti-inflammatory drugs (NSAIDs) that block the activity of COX2 could contribute to the mitigation of cancer-linked mortality [8]. For example, Harris et al. published that celecoxib and rofecoxib effectively reduced the risk of development of colon cancer in a study conducted with the enrolment of 326 patients with colon cancer of invasive type and 652 disease naïve control subjects [9]. Added to that, the feasibility of COX2 inhibitor NSAIDs in tumor growth prevention and the lengthening of survival were further confirmed at preclinical level [10]. 

COX2 together with the microsomal prostaglandin E2 synthase-1 (mPGES-1) are the two major contributors to PGE2 synthesis through the prostanoid biosynthetic pathway [1]. In addition, PGE2 concentration is also influenced by the activity of 15-hydroxyprostaglandin dehydrogenase (15-PGDH)—*the main catalysator of the oxidisation of PGE2 to inert 15-keto-PGE2* [11,12]. Beyond the synthesis processes and the degradation of PGE2, both the intra- and the intercellular PGE2 levels are mediated by drug resistance-associated protein (MRP4) and prostaglandin uptake transporter (PGT) [13,14].

Identically to COX2, there is a notable interplay between the excess production of PGE2 and tumorigenesis [15]. PGE2 acting via G-protein coupled rhodopsin-type prostanoid receptors E 1–4 (EP1, EP2, EP3, EP4) triggers various tumor-associated intracellular signaling pathways, out of which the most remarkable ones are the stimulation of the epidermal growth factor-receptor (EGFR), the phosphorylation of protein kinase C (PKC)-mediated extracellular signal-regulated kinase (ERK), and the activation of G-protein dependent beta-catenin [1,2,12,16,17]. The PGE2 regulated activation of EP receptor induces tumor-related angiogenesis along with the enhancement of tumor cell invasion and the halting of natural immune responses [12]. Further, PGE2-generated tumor proliferation as well as apoptosis inhibition lead to tumor volume expansion [12]. Former translational studies registered moderate tumor propagation in EP2-knockout mice in comparison with the wild-type ones [18]. Recent research indicating a link between 15-PGDH down-regulation and the appearance of certain malignancies including lung and transitional bladder cancers may project the role of this degrading enzyme in carcinogenesis [19,20]. Moreover, the metastatic spread of triple negative breast cancer was promoted by MRP4 [21]. In addition, *Tong and colleagues* claimed that PGE2 may serve as a biomarker in different malignancies such as pancreatic, breast, oral, or renal cancer [1].

Target-specific molecules directed towards the components of the COX2/PGE2/EP axis seem to be promising imaging probes in the early diagnostics of PGE2pos. neoplastic alterations as well as in the design of anti-cancer drug candidates. Nuclear medical in vivo preclinical model systems applying target-selective radiolabelled compounds may aid to further understand the molecular pathways of COX2/PGE2 that would attribute to the identification of well-suited diagnostic and anti-tumor biomarkers. 

In the present review we provide a detailed overview about PGE2- specific radiolabelled diagnostic probes and their potential therapeutic applications in PGE2 overexpressing malignancies. 

### 1.2. PGE2 and Cyclodextrins (CDs)

Cyclodextrins (CDs) are arising as powerful diagnostic agents in the field of nuclear medicine. Built up by D-(+)-glucopyranose molecules, these truncated cone-shaped oligosaccharides have a hydrophobic inner cage and a hydrophilic outer surface [22,23,24,25]. Given their advantageous physicochemical characteristic features, CDs are highly valuable for forming inclusion complexes with a myriad of molecules including PGE2. The complex forming capability of β-CDs with PGE2 was already established by Hirayama et al. in 1984 [26]. Exploiting the beneficial biochemical characteristics of β-CDs, *Hirayama and co-workers* managed to obtain the stabilization of PGE2 in aqueous solution with the application of the following two modified β-CD derivatives: heptakis (2, 6-di-O-methyl)-β-CD (DM-β-CD) and heptakis (2, 3, 6-tri-O-methyl)-β-CD (TM-β-CD). Compared to TM-β-CD, DM-β-CD appeared to be more effective in the maintenance of the stability of PGE2 projecting its feasibility in the proposal of pharmaceutical dosage formulas [26]. 

Besides the above-mentioned CD compounds, the complexation of another β-CD derivative—*randomly methylated β-cyclodextrin (RAMEB)*—with PGE2 was exhaustively evaluated as well. In a recent in vitro study focusing on the interaction between RAMEB and various proinflammatory molecules such as PGE2, substance P (SP), bradykinin (BRK), and calcitonin gene-related peptide (CGRP), *Sauer and co-workers* observed that PGE2 displays more significant binding potential to RAMEB in comparison with the other investigated agents [27]. To indicate the best fitting β-CD guest molecule, the Molecular Operating Environment docking algorithm-based molecular docking of SP, CGRP, BRK, and PGE2 was carried out. While RAMEB-affine PGE2 (ΔGbind= −12.57 kcal/moL) was characterised by high CD binding capability, BRK was moderately connected to RAMEB (−10.54 kcal/moL). Neither of the remaining assessed molecules showed affinity towards RAMEB. Earlier studies also focused on the evaluation of the complexation between CDs and other PGE subtypes, such as prostaglandin E1 (PGE1). Inoue et al. managed to obtain the enhanced stability of PGE1 derivative Limaprost/α-CD complex (Limaprost-alfadex) in humid circumstances by adding β-CD to the existing agent [28]. Later, they investigated the way of complexation between the PGE1 compound and the CD molecules, applying ^1^H- and ^13^C-Nuclear Magnetic Resonance (NMR) spectroscopy [29]. Referring to their results, a kind of ternary complex formation could be accountable for the stabilization of the drug. Overall, their findings also strengthened the strong association between CDs and PGE molecules. 

Based on the above remarked former discoveries, PGE2-affine CDs appear to be valuable diagnostic vectors and drug carrier vehicles in PGE2-associated pathological processes, including carcinogenesis. Moreover, radiolabelled CDs complexed with PGE2 may establish the basis of target-specific diagnostics of PGE2 expressing malignancies. Applying radiometals with therapeutic radiation for labelling purposes, the theranostic potential of PGE2-directed CDs could also be tested (Figure 1). Prior to the definitive integration of PGE2-affine CD-based imaging probes into routine clinical use, the need for the verification of their in vivo diagnostic applicability is warranted. Preclinical small animal model systems ensure appropriate circumstances for the assessment of the stability, the pharmacokinetics, the biodistribution, and the tumor-homing effectiveness of radiolabelled CD compounds.

Positron emission tomography (PET) is the mainstay diagnostic device in the detection of primary tumors and pertinent metastases. PET purveys data on in vivo physiological and pathological processes with the application of various targeted radiopharmaceuticals containing positron emitting radioisotopes [30]. The non-invasive characteristics and high sensitivity of PET imaging make it well-suited for the preclinical evaluation of tumor-selective PGE2-specific molecules [31,32]. 

In 2019, *Hajdu and his colleagues* launched a novel Gallium-68 (^68^Ga)-labelled, p-NCS-benzyl-NODA-GA (NODAGA)-conjugated β-CD compound—denoted as ^68^Ga-NODAGA-(2-hydroxypropyl)-β-cyclodextrin ([^68^Ga]Ga-NODAGA-HPBCD)—to verify its physicochemical adequacy for potential future in vivo PET diagnostics [31]. This research could be hailed as a landmark study in the proposal of further radiolabelled CD-based derivatives for diagnostic purposes. In their study they elaborated the radiochemical synthesis process of this newly constructed PET probe. The exceptional radiochemical purity (RCP; 96–99%) (defined by radio-HPLC), the hydrophilic physicochemical property (*LogP* = −3.07 ± 0.11), and the stability in mouse serum confirmed the feasibility of [^68^Ga]Ga-NODAGA-HPBCD in upcoming in vivo investigations. Applying healthy 12-week-old male BALB/c mice in vivo PET/CT examinations indicated a renal way of excretion along with low tracer uptake values measured 90 min postinjection in the liver (SUV_mean_: 0.12 ± 0.03), the intestines (SUV_mean_: 0.11 ± 0.02), the heart (SUV_mean_: 0.13 ± 0.02), the lung (SUV_mean_: 0.17 ± 0.02), and the brain (SUV_mean_: 0.09 ± 0.02). The ex vivo biodistribution data were in line with the in vivo figures. Since the established synthesis process led to the production of an intravenously (iv.) applicable radiotracer with adequate organ distribution and outstanding chemical features, it seemed to be adaptable for the manufacturing of additional radio-conjugated CD molecules. Hence, CD-based, PGE2-specific PET radiopharmaceuticals (Figure 2) could be established for cancer imaging on the basis of the synthesis model of Hajdu et al. Table 1. displays the in vivo preclinical studies with PGE2-affine radiolabelled imaging probes. 

### 1.3. PGE2pos. Preclinical Tumor Models

BxPC-3 pancreatic adenocarcinoma cell lines are firmly associated with elevated PGE2 synthesis and EP2 receptor upregulation [36]. Investigating the following pancreatic cell lines in vitro: MiaPaCa-2, BxPC-3, PANC-1, and Capan-1, *Takahashi and co-workers* observed the expression of EP and various growth factor receptors by reverse transcriptase-polymerase chain reaction (RT-PCR). EIA-based (Enzyme immunoassay) determination of PGE2 levels and the presence of COX1 and COX2 mRNA were also evaluated. Applying EIA, ten-fold higher PGE2 concentrations were observed in BxPC-3 culture media relative to the other investigated specimens. Moderate EP2 mRNA presence and enhanced EP4 mRNA expression were identified in BxPC-3 cells. Further, comparing all investigated cells lines, the most remarkable COX- and EP4-reliant autocrine loop was manifested in the BxPC-3 culture. 

The discrete presence of COX2 enzyme and related inconsiderable PGE2 synthesis of Panc-Tu-1 pancreatic ductal adenocarcinoma (PDAC) cells was reported by Gonnermann et al. [37]. Four PDAC cells lines including PancTu-1, Pt45P1, Panc89, and Colo357 were analysed in their study. As part of the comparison analyses of the PDAC cell cultures, disproportions between the appearance of COX2 and COX2-generated PGE2 synthesis were determined. In accordance with the PGE2 levels, in PancTu-1 and in Panc89 cells both flow-cytometry- and Western Blot-based measurements revealed moderate COX2 presence in comparison with receptor upregulated Colo357 cell lines. Less than 0.5 ng/mL PGE2 was produced by PancTu-1 and Panc89 cells, while Colo357-associated synthesis exceeded 6 ng/mL. In a like manner, Szabó et al. also confirmed the low EP2 receptorial expression of PancTu-1 cells during the immunohistochemical verification of the presence of EP2 receptor in several cell lines [35]. 

To our best knowledge, no exact literature data is available regarding the EP2 receptor status of HT1080 cells; however, according to the results of some earlier studies, the existence of these receptors and pertinent PGE2 production may be concluded. Anti-Prostaglandin E Receptor EP2 antibody immunohistochemistry with 3,3-diaminobenzidine (DAB) chromogens was employed to detect the appearance of EP2 receptor on HT1080 cells [35]. Histological staining revealed low amount of membrane EP2 receptors on the cell surface of HT1080 cell lines. Pifithrin-α (PFT-α)-induced COX2 level enhancement experienced in HT1080 cells with p53 wild-type suggests the presence of both PGE2 and its receptors in fibrosarcoma cell lines [38]. P-53 transactivation inhibitor PTF-α triggers COX2 overexpression via mitogen-activated protein kinase (MAPK) kinase (MEK)/extracellular signal-regulated kinase (ERK)-based molecular mechanisms. In a bid to investigate PFT-α-generated COX2 overexpression, wild-type p53 HT1080 fibrosarcoma cells were administered with 40 µmol/L PFT-α. COX2 presence was authenticated applying Western blot analyses. Compared to the control, HT1080 cells exhibited 5.3-fold higher COX2 presence after PFT-α application. 

Pi et al. established an anti-programmed cell death protein 1 (anti-PD-1) drug-resistant B16-F10 (B16-F10-R) preclinical melanoma *Pdcd1* transgenic mouse model system to investigate the expression of programmed death-ligand 1 (PD-L1) and lymphocyte invasion into the tumor niche [39]. Additionally, associations between COX2 and anti-PD-1 pembrolizumab resistance were surveyed as well. At an average tumor bulk of 50 mm^3^, beyond the iv. administration of 10 mg/BW (kg) pembrolizumab (anti-PD-1 antibody) or its control type twice a week, aspirin (10 mg/BW), SC560 (selective COX1 inhibitor, 5 mg/BW), celecoxib (selective COX2 inhibitor, 5 mg/BW), and E7046 (EP4 inhibitor, 10 mg/BW) were intraperitoneally injected three times weekly to the following tumor-bearing experimental small animal groups: B16-F10, B16-F10-NR (anti-PD-1 non-resistant), B16-F10-R (anti-PD-1 resistant), and B16-F10-R-knockout ptgs2 (B16-F10-knockout prostaglandin endoperoxide synthases/COX2). Prostaglandin endoperoxide synthases 1/prostaglandin endoperoxide synthases2 (Ptgs1/Ptgs2, COX1/COX2) transcript levels and PGE2 values were determined applying Real-Time Quantitative Polymerase Chain Reaction (RT-qPCR) and Enzyme-Linked Immunosorbent Assay (ELISA). Amongst others, anti-COX2 antibody was utilized for the accomplishment of Western Blot analyses, while T and natural killer (NK) cells were identified by flow cytometry-based staining. B10-F10-R malignancies were featured with enhanced PD-L1 expression-associated reduced CD3^+^, CD4^+^, and CD8^+^ T cell and NK cell invasion, as well as the overexpression of the *ptgs2* gene. Albeit these phenomena seemed to be reversible with the co-administration of non-selective COX1/COX2 inhibitor and anti-PD-1 antibody. Upregulated COX2 levels with associated PGE2 synthesis characterised the B16-F10-R tumor-bearing mice, reflecting the major contribution of COX2 to anti-cancer drug sensitivity. Further, anti-tumor drug insensitivity was abolished utilizing *ptgs2* knockout or E7046 selective EP4 inhibitor combined with anti-PD-1 treatment. Given the COX2 axis inhibition-originated heightened immune cell invasion (T and NK cells) as well as therapeutic drug-sensitivity we presume the role of EP2 receptors and PGE2 in the biological processes of B16-F10 cell lines.

B-cell-associated immunoglobulin production is regulated by PGE2 in immunological-allergic processes [40]. Of note, 9 times higher serum IgE concentrations and much elevated monocyte-driven PGE2 synthesis characterised the patients suffering from progressive Hodgkin’s disease (HD). Previous literature data confirm the PGE2-related EP receptor positivity of B lymphocytes [41,42,43]. Further, in vivo investigation of the receptor profile of mouse B lymphocytes revealed the presence of type 2 and type 4 EP receptors on the surface of the cell membrane (EP2, EP4) [44]. Taking the B-cell origin of HD into consideration, these findings may imply the strong EP receptor expression of B cells and different B cell lines. 

More recently, chemically-induced, rat-originated syngeneic Ne/De (rat mesoblastic nephroma) and He/De (rat hepatocellular carcinoma) tumors and their metastases also proved to be PGE2pos. during preclinical PET imaging studies [35].

## 2. Insight into Preclinical In Vivo Nuclear Medical Studies

Driven by prior literature findings and the radiochemical results of *Hajdu and co-workers,* Trencsényi et al. proposed a novel PGE2-selective radiotracer—[^68^Ga]Ga-NODAGA-RAMEB—to assess its tumor-targeting competence as well as its organ distribution at the preclinical level, applying PET [33]. To accomplish in vivo and ex vivo experiments, PGE2 overexpressing BxPC-3 human pancreas tumors and PancTu-1 human pancreas adenocarcinoma with low receptorial expression were subcutaneously (sc.) generated in CB17 severe combined immunodeficient (SCID) mice. Based on the receptor profile of the selected tumors, they were appropriate for the conduction of this research. Former evidence implied excess production of PGE2 in BxPC-3 cells [45]. In reference to flow cytometry and Western Blot analysis results, Gonnermann et al. observed insignificant COX2 presence and PGE2 synthesis in Panc-Tu-1 cell lines [37]. In a similar vein, *Trencsényi and his team* verified marked EP2 receptor expression in BxPc-3 tumors, while PancTu-1 pancreatic neoplasms were depicted with moderate receptorial presence using immunohistochemical staining (Figure 3A). With the application of MiniPET-II small animal PET scanner, postinjection of 7.3 ± 0.3 MBq [^68^Ga]Ga-NODAGA-RAMEB through the lateral tail vein, in vivo static and dynamic PET acquisition took place under inhalation anaesthesia in both control and in tumor-bearing (BxPC-3 and PancTu-1) small animals to analyse in vivo tracer distribution. For the objective evaluation of the tracer accretion of the tumor, different organs, and tissues, *Trencsényi and co-workers* determined the subsequent quantitative parameters: standardized uptake value (SUV) and tumor-to-muscle ratio (T/M). In the context of the ex vivo measurements, 30, 60, and 90 min—followed by the iv. application of 7.31 ± 0.32 MBq [^68^Ga]Ga-NODAGA-RAMEB—tumor naïve and tumorous animals were put into sleep to remove three tissue samples for radioactivity (presented in %ID/g tissue) determination. Regarding healthy mice, 90 min post-administration, quantitative SUV_mean_ figures—*corresponding to the qualitative assessment*—revealed low tracer uptake in the lung (SUV_mean_: 0.31 ± 0.07), heart (SUV_mean_: 0.12 ± 0.04), intestines (SUV_mean_: 0.13 ± 0.03), liver (SUV_mean_: 0.16 ± 0.06), stomach (SUV_mean_: 0.12 ± 0.04), and in the brain (SUV_mean_: 0.12 ± 0.04), whereas, due to the renal excretion, elevated urine radioactivity was registered. The in vivo observed urinary route of elimination was in accordance with both the ex vivo data and the results of Hajdu et al. with [^68^Ga]Ga-NODAGA-HPBCD [31,33]. Referring to the outcomes of other prior research studies, quick renal clearance was disclosed both at clinical and preclinical levels using either radio-appended or non-radiolabelled HPBCD [31,46,47]. The quantitative in vivo parameters corresponded to the ex vivo ones measured by a calibrated gamma counter (Perkin-Elmer Packard Cobra, Waltham, MA, USA). Although no disparities were recorded in case of the tracer accumulation of the urine, kidneys, intestines, fat tissue, lung, and brain 30, 60, and 90 min postinjection, notable unevenness was disclosed between the 30 min and the 90 min %ID/g figures of the blood, small intestine, stomach, muscle tissue, heart, bone, salivary glands, gall bladder, and the pancreas *(p* ≤ 0.01*).* Of note, enhanced [^68^Ga]Ga-NODAGA-RAMEB uptake of the lungs could be explained by the prolonged presence of the tracer in the watery pulmonary areas (%ID/g: 0.28 ± 0.08, 0.21 ± 0.04, 0.14 ± 0.03, 30, 60, and 90 min postinjection, respectively). Comparing the ID%/g and SUV values of [^68^Ga]Ga-NODAGA-RAMEB and [^68^Ga]Ga-NODAGA-HPBCD, [^68^Ga]Ga-NODAGA-RAMEB accumulation was more moderate in the thoracic and abdominal organs [31,33]. The assessment of the in vivo PET images of the BxPC-3 tumor-bearing mice revealed that—*owing to the reduction in the background activity*—the contrast of the tumors was directly proportional to the length of the incubation time (T/M SUV_mean_: 7.80 ± 1.64 and T/M SUV_mean_: 18.57 ± 2.64, 10 and 90 min after the tracer injection, respectively). Thanks to this noteworthy T/M ratio (highest at 80–90 min post-radiopharmaceutical administration), PET images with superior quality/contrast could be ensured in PGE2 upregulated BxPC-3 tumorous experimental animals. Supported by the 3–14-fold lower SUV_mean_, SUV_max_, T/M SUV_mean_, and T/M SUV_max_ values of 0.04 ± 0.01, 0.08 ± 0.02, 1.33 ± 0.19 and 1.66 ± 0.22, respectively, PancTu-1 tumors with discrete receptor expression could only be characterised by faint [^68^Ga]Ga-NODAGA-RAMEB uptake. These measures are in line with the ex vivo data confirming 5-times lessened radiolabelled CD uptake of the PancTu-1 neoplasms 90 min after [^68^Ga]Ga-NODAGA-RAMEB administration. Within the framework of the ex vivo studies *Trencsényi and colleagues* experienced apparently elevated (*p* ≤ 0.01) radiotracer uptake and T/M ratios in case of the BxPC-3 tumors compared to the other organs. This phenomenon reflects the outstanding PGE2 binding potential of [^68^Ga]Ga-NODAGA-RAMEB. Based on the aforementioned in vivo qualitative PET results as well as the ex vivo gamma counter findings, Trencsényi et al. successfully corroborated—*for the first time*—the PGE2 specificity of the newly launched [^68^Ga]Ga-NODAGA-RAMEB PET probe. Consequently, their study represents a breakthrough in the establishment of the basis of PGE2-specific in vivo PET imaging. 

Initiated by the findings of *Trencsényi and colleagues*, Csige et al. presented another PGE2-affine RAMEB compound with potential future diagnostic as well as therapeutic applications [34]. In this study the same β-CD—*RAMEB*—was conjugated with chelator DOTAGA and radiolabelled with both ^68^Ga and Bismuth-205/206 (^205/206^Bi; the surrogate of 213-Bismuth/^213^Bi) [34]. Taking the perfect complexation characteristics and kinetic inertness of DOTAGA into account, it seemed ideal to be attached to the amino group of NH2-RAMEB [48,49]. The whole process of the biosynthesis as well as the in vivo distribution and pharmacokinetic profile of both ^68^Ga and ^205/206^Bi-appended DOTAGA-RAMEB were outlined in detail. For this purpose, in vivo PET examinations and ex vivo biodistribution experiments were brought about, applying CB17 SCID male PGE2 receptor positive BxPC-3 tumor-bearing mice of 12-weeks of age. The enhanced EP2 receptorial expression of the BxPC-3 tumors was immunohistochemically strengthened in the cell plasm and in the cell membrane as well. Approximately 12 days post tumor cell inoculation, at a tumor volume of 98 ± 5 mm^3^, followed by the iv. injection of 6.4 ± 0.3 MBq [^68^Ga]Ga-DOTAGA-RAMEB, in vivo static and dynamic PET imaging was performed. Healthy experimental mice—*as control cohort*—also underwent PET acquisition. As part of quantitative PET data analysis, SUV values and T/M ratios were registered. In line with the hydrophilic nature of [^68^Ga]Ga-DOTAGA-RAMEB *(LogP:-3.5)*, markedly enhanced tracer uptake was pinpointed in the kidneys and in the urinary bladder of the healthy control mice. In a like manner—*as remarked above*—Trencsényi et al. also observed renal excretion in case of [^68^Ga]Ga-NODAGA-RAMEB [33]. In contrast, the qualitative PET image assessment of the healthy mice revealed slight radiopharmaceutical accumulation in the thoracic (lungs, heart) and abdominal organs (intestines, liver, stomach). These results of Csige et al. correspond to those of *Trencsényi and co-workers* [33,34]. Consequently, we can conclude that the change in the chelator did not influence tracer uptake. In addition, this newly synthesized ^68^Ga-labelled, DOTAGA-chelated PET probe made the visualization of BxPC-3 tumors with PGE2 upregulation possible with the highest T/M ratios defined at 90 min post tracer application (2.5 ± 0.2) (Figure 3B). This way the PGE2pos. tumor-targeting potential of radiolabelled RAMEB—already established by Trencsényi et al.—was further authenticated. Identically, Trencsényi et al. also noted the most elevated T/M ratios 80–90 min following iv. tracer administration [33]. This finding further strengthens that the PGE2-binding affinity of ^68^Ga-labelled RAMEB is regardless of the applied chelator. 

Beyond the detailed in vivo investigations, ex vivo studies were also executed to capture the totality of the organ distribution profile of both [^68^Ga]Ga-DOTAGA-RAMEB and [^205/206^Bi]Bi-DOTAGA-RAMEB. Followed by the injection of the tumor-bearing and healthy mice with 6.2 ± 0.2 MBq or 0.86 ± 0.17 MBq [^68^Ga]Ga-DOTAGA-RAMEB, or [^205/206^Bi]Bi-DOTAGA-RAMEB, respectively, different organ samples were extracted from the experimental animals at pre-defined time points (30, 60, and 90 min postinjection) for radioactivity determination. No disproportion was pinpointed between the %ID/g values of either ^68^Ga or ^205/206^Bi-labelled DOTAGA-RAMEB at the 30 or 60 min time points. Neither did we find any differences between the tracer uptake of the BxPC-3 tumors at either investigated time points. Similarly to the earlier studies with [^68^Ga]Ga-NODAGA-RAMEB and [^68^Ga]Ga-NODAGA-HPBCD, the selected organs and tissues were depicted with moderate [^68^Ga]Ga-DOTAGA-RAMEB accumulation 90 min post-administration [31,33]. On the contrary, the spleen, the colon, the stomach and the adipose tissue showed remarkable [^205/206^Bi]Bi-DOTAGA-RAMEB concentration. 

Featured with RCP above 98%, and satisfactory in vitro and in vivo stability, both [^68^Ga]Ga-DOTAGA-RAMEB and [^205/206^Bi]Bi-DOTAGA-RAMEB seem to be promising PET probes in the accomplishment of forthcoming preclinical research. Besides their outstanding diagnostic feasibility in PGE2pos. tumor detection, DOTAGA-RAMEB conjugation with therapeutic radiometals may broaden the horizons of theranostical applications. 

Considerable attention centered around radiolabelled RAMEB and HPBCD has spawned novel studies focusing on the evaluation of their diagnostic efficacy in other PGE2pos. tumor types besides BxPC-3. The tumor-targeting competence of RAMEB and HPBCD complexed with ^68^Ga-NODAGA was assessed in vivo in HT1080 (human fibrosarcoma), A20 (mouse B cell lymphoma), PancTu-1 (human pancreas adenocarcinoma), BxPC-3 (human pancreas adenocarcinoma), B16-F10 (mouse melanoma), Ne/De, and He/De tumor-bearing model systems (Figure 3C) [35]. Immunohistochemistry-based receptor profile analysis disclosed high EP2 receptor density in A20, BxPC-3, B16-F10, Ne/De, and He/De tumors, whereas HT1080 and PancTu-1 neoplasms were featured with the faint presence of EP2 receptors. For tumor induction, male CB17 SCID mice of 12 weeks of age were sc. transplanted with H1080, A20, PancTu-1, BxPC-3, B16-F10, Ne/De, and He/De cell lines. Further, subrenal capsule assay (SRCA) was performed to generate He/De tumors under the left renal capsule of 16-week-old female Fischer-344 rats. Ten MBq [^68^Ga]Ga-NODAGA-RAMEB or [^68^Ga]Ga-NODAGA-HPBCD was given to all experimental animals for the accomplishment of in vivo static PET images in order to study the biodistribution of the investigated PET probes in all neoplasms. Supplementary [^18^F]-Fluorodeoxyglucose ([^18^F]F-FDG) PET images—*performed to verify the presence of the tumors*—revealed no [^18^F]F-FDGpos. regions in case of the HT1080, PancTu-1, and BxPC-3 tumors. Minor tumor volume with related decreased cell number as well as modest glucose usage and metabolism may underpin this finding. Despite the measurable difference between the amount of tracer uptake of the two ^68^Ga-labelled CD derivatives, both of them seemed to be feasible in the detection of the experimental neoplasms. We suppose that the variability in EP2 receptor expression and PGE2 levels between the HT1080, A20, and B16-F10 tumors may explain why the concentration of the two molecular probes considerably differed. Upon PET image analyses, the extent of PGE2 positivity exceeded that registered on the histological specimens. Szabó et al. hypothesized that alterations in the receptor status during tumor development as well as the binding of the CD probes to both the EP2 receptor and the PGE2 itself lead to more outstanding receptor positivity on the PET scans.

Tumor cells developing under the left renal capsule of the experimental animals give metastasis to the parathymic lymph node (PTLN) of the thorax. Although the primary Ne/De tumors and their PTLN metastases accumulated [^18^F]FDG and both ^68^Ga-labelled agents (Figure 4), [^18^F]FDG uptake was two times higher compared to the accumulation of the CD derivatives *(p* ≤ 0.01*)*. In addition, the primary Ne/De tumors were visualized using both ^68^Ga-labelled imaging probes; however, more increased SUV values were detected in case of the HPBCD derivative (SUV_mean_: 3.52 ± 0.23; SUV_max_: 4.80 ± 0.21) relative to the RAMEB compound (SUV_mean_: 2.51 ± 0.19; SUV_max_: 3.21 ± 0.35). A similar uptake pattern was registered in case of the metastatic PTLNs that showed double [^68^Ga]Ga-NODAGA-HPBCD accumulation in comparison with the RAMEB agent. Despite the fact that in case of the primary malignancies no considerable difference was registered between the T/M ratios of [^18^F]FDG and the radiolabelled CD probes (*p* ≤ 0.05), as for the PTLN metastases markedly more enhanced tumor-to-non-target ratios were depicted with [^18^F]FDG in comparison with the T/M ratios of the CD-based radiopharmaceuticals *(p* ≤ 0.01*).* In accordance with the immunohistochemical findings that confirmed lower PGE2 receptor presence in the PTLN metastases, the secondary neoplasms demonstrated 3–5-times more decreased accumulation of the radiolabelled CD derivatives compared to the primary Ne/De tumors.

Comparing the radiotracer accretion of the SRCA-induced and the sc. transplanted Ne/De tumors, more increased accumulation of the radiolabelled CDs was depicted in the case of the subrenally growing Ne/De tumors compared to the sc. developing ones. This could probably be attributed to the orthotopic subrenal biological media that favours tumor propagation and metastatic spread [50]. In accordance with this finding, *Máté and co-workers* experienced the same during the comparison of the presence of neoangiogenesis-related molecules between SRCA-generated and sc. transplanted Ne/De tumor models [51]. Gamma counter-based ex vivo radioactivity measurements—*in line with the* in vivo *data*—demonstrated that PGE2 enriched BxPC-3, A20, Ne/De and He/De tumors exhibited the most elevated uptakes of both ^68^Ga-labelled CDs. Although, Szabó et al. remarked a notable difference between the ex vivo tracer accumulation of the SRCA-induced Ne/De neoplasms and the sc. growing Ne/De tumors. In summary, the research findings of *Szabó and colleagues* further confirm that the place for radiolabelled CDs—*RAMEB and HPBCD*—is warranted in the molecular diagnostics of PGE2 expressing tumors. 

## 3. Closing Remarks

Although recent years have witnessed milestone advances in battling against cancer, there are still several uncovered fields in tumorigenesis that remain to be better elucidated. Therefore, growing interest has been placed upon the investigation of tumor formation and progression at a molecular level. A broad set of experiments has intensively evaluated the role of RAS oncogene—*the member of small GTPases*—in cancer development [52]. GTP-activated H/N/K-RAS stimulates a large number of down-stream signalling pathways including Raf/MAPK kinase, MEK/ERKs, and PI3K/Akt that exert regulatory effects on cell proliferation, survival, apoptosis, and differentiation [52,53,54,55]. RAS mutations with related steady oncogenic activity can be encountered in a wide range of malignancies [53,56]. Beyond RAS-mediated signalling actions, COX2 and COX2-driven PGE2 production also have central contribution to neoplastic transformation. Based on literature data, the moderation of cancer-linked COX2 expression and COX2-triggered PGE2 synthesis is strongly associated with the activity of RAS oncoprotein [57,58,59]. On the other hand, COX2-dependent PGE2 production upregulates RAS signalling [53].

Taking the outstanding role of COX2/PGE2/RAS feedback circuit in carcinogenesis into account, uncovering the entire molecular pathway may favour our understanding of the underlying mechanisms of tumor development and metastatic spread. Broadened knowledge on the functioning of the COX2/PGE2/RAS axis might be for the benefit of the proposal of novel diagnostic molecular probes as well as anti-tumor drug candidates that could lead to the ultimate goal of the establishment of tailor-made cancer treatment.

## Figures and Tables

**Figure 1 ijms-24-06942-f001:**
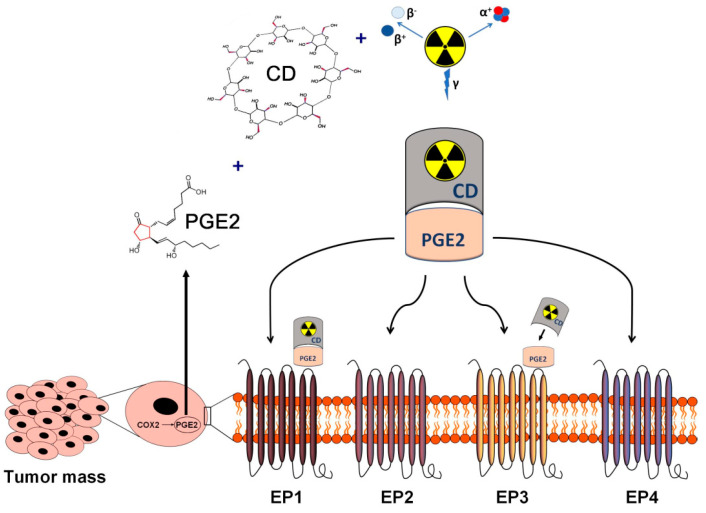
Schematic overview of the in vivo diagnostics and radioisotope therapy of PGE2-expressing and PGE receptor-positive tumors using radiolabelled CDs. Applying radioisotopes with diagnostic (γ) and therapeutic radiation (α or β), radiolabelled CDs complexed with tumor cell derived PGE2 may establish the basis of target-specific diagnostics and personalized radionuclide therapy of PGE2-expressing malignancies. CD: cyclodextrin; PGE2: prostaglandin E2.

**Figure 2 ijms-24-06942-f002:**
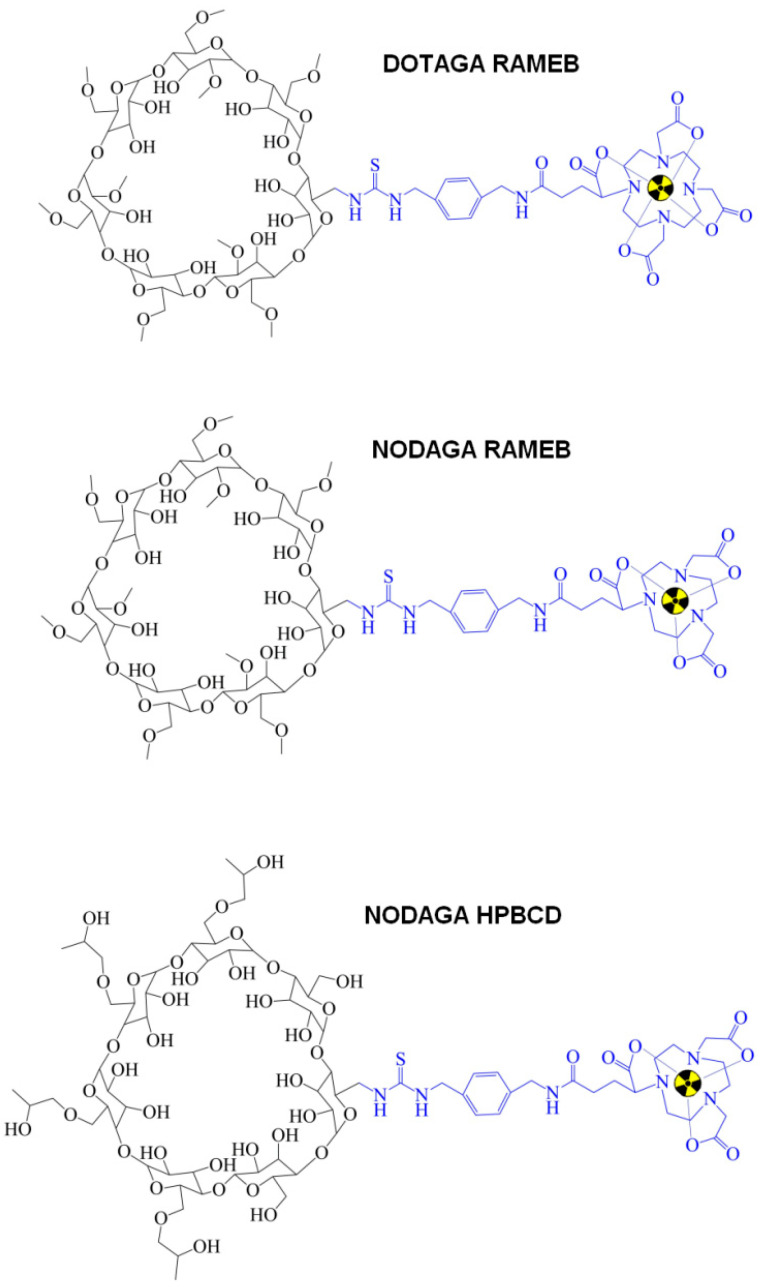
Chemical structure of CD-based PET radiopharmaceuticals. CD: cyclodextrin; PET: positron emission tomography.

**Figure 3 ijms-24-06942-f003:**
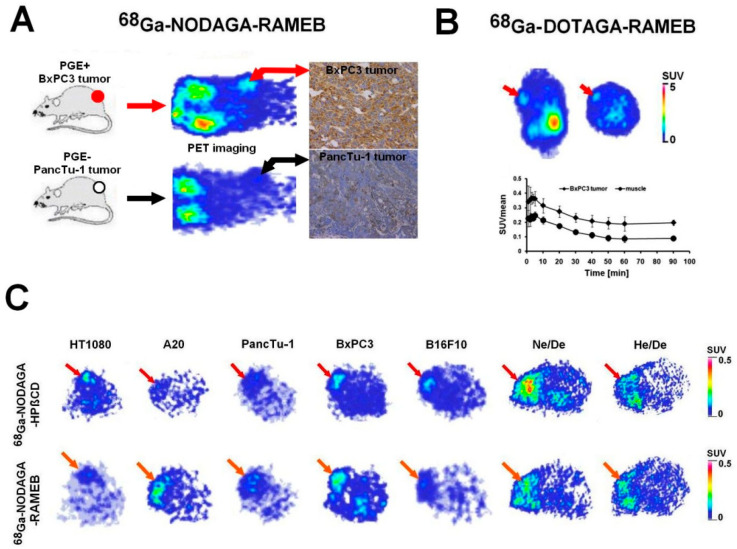
In vivo PET imaging of experimental tumors using ^68^Ga-labelled CDs. (**A**) panel: In vivo assessment of [^68^Ga]Ga-NODAGA-RAMEB accumulation in PGE2 positive (BxPC3) and negative (PancTu-1) human pancreas adenocarcinoma tumors using miniPET imaging 90 min after the intravenous injection of [^68^Ga]Ga-NODAGA-RAMEB. Decay-corrected PET images were obtained 12 ± 1 days after subcutaneous cancer cell inoculation. Red arrows: BxPC3 tumors; black arrows: PancTu-1 tumors. From [33] with permission. The microscope magnification was 40×. (**B**) panel: In vivo accumulation of the intravenously injected [^68^Ga]Ga-DOTAGA-RAMEB in BxPC-3 tumor in SCID mice. Representative decay-corrected static coronal (left) and transaxial (right) PET images 90 min after radiotracer injection. Time-activity curve of [^68^Ga]Ga-DOTAGA-RAMEB in BxPC-3 tumor and muscle (background). Red arrows: subcutaneously growing BxPC-3 tumor. From [34] with permission. (**C**) panel: Representative decay-corrected transaxial PET images of experimental tumors after the intravenous injection of [^68^Ga]Ga-NODAGA-HPBCD/HPβCD (red arrows), [^68^Ga]Ga-NODAGA-RAMEB (orange arrows). Decay-corrected PET images were obtained 10 ± 2 days after subcutaneous tumor cell injection and 90 min after the intravenous injection of the ^68^Ga-labelled CDs. From [35] with permission. A20: mouse B cell lymphoma; BxPC-3: human pancreas adenocarcinoma; B16-F10: mouse melanoma; CD: cyclodextrin; DOTAGA: 1,4,7,10-tetrakis(carboxymethyl)-1,4,7,10-tetraazacyclododecane glutaric acid; ^68^Ga: Gallium-68; He/De: rat hepatocellular carcinoma; HPBCD/HPβCD: hydroxypropyl-β-cyclodextrin; HT1080: human fibrosarcoma; Ne/De: rat mesoblastic nephroma; NODAGA: 1,4,7-triazacyclononane-1-glutaric acid-4,7-diacetic acid; PancTu-1: pancreas ductal epitheloid carcinoma; PET: positron emission tomography; PGE2: prostaglandin E2; SCID: severe combined immunodeficient; SUV: standardized uptake value; RAMEB: randomly methylated beta-cyclodextrin.

**Figure 4 ijms-24-06942-f004:**
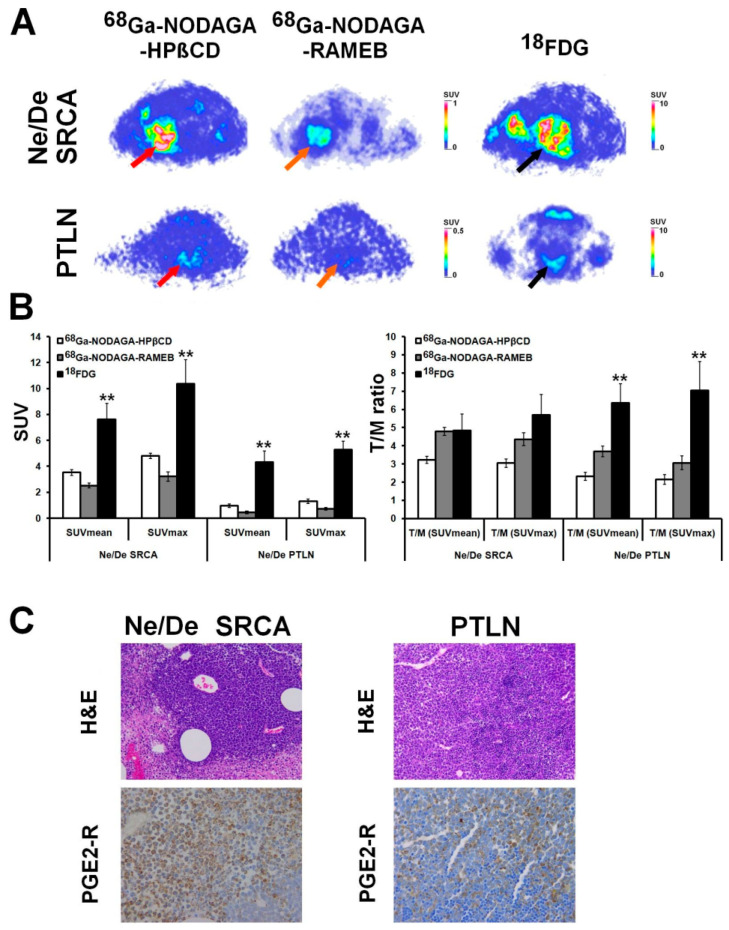
PET imaging of primary tumors and metastases using ^68^Ga-labelled CDs. (**A**) panel: In vivo assessment of ^68^Ga-labelled CDs ([^68^Ga]Ga-NODAGA-HPBCD/ HPβCD—red arrows; [^68^Ga]Ga-NODAGA-RAMEB—orange arrows), and [^18^F]F-FDG (black arrows) accumulation of SRCA-induced Ne/De primary tumors and parathymic lymph node (PTLN) metastases using preclinical PET imaging. (**B**) panel: Quantitative SUV data analysis of the PET images. PET images and SUV data were obtained 8 days after the SRCA, and 50 and 90 min after the intravenous injection of [^18^F]F-FDG and the ^68^Ga-labelled probes, respectively. SUV values are presented as mean ± SD. (**C**) panel: Histological analysis of Ne/De primary tumors (SRCA) and PTLN metastasis. Upper row: hematoxylin-eosin (H&E) stained tumor tissue. Magnification: 20×. Lower row: Anti-Prostaglandin E Receptor EP2/PTGER2 antibody immunohistochemistry (PGE2-R), visualized with 3,3-diaminobenzidine (DAB) (brown staining). Magnification: 40×. Figures are from [35] with permission. CD: cyclodextrin; DAB: 3,3-diaminobenzidine; [^18^F]F-FDG: Fluorine-18 labelled Fluorodeoxyglucose; ^68^Ga: Gallium-68; NODAGA: 1,4,7-triazacyclononane-1-glutaric acid-4,7-diacetic acid; H&E: hematoxylin-eosin; HPBCD/HPβCD: hydroxypropyl-β-cyclodextrin; Ne/De: rat mesoblastic nephroma; PET: positron emission tomography; PGE2-R: Anti-Prostaglandin E Receptor EP2/PTGER2 antibody immunohistochemistry; PTLN: parathymic lymph node; RAMEB: randomly methylated beta-cyclodextrin; SRCA: subrenal capsule assay; SUV: standardized uptake value; SD: standardized uptake value. ** *p* < 0.01.

**Table 1 ijms-24-06942-t001:** Overview of preclinical studies with radiolabelled prostaglandin E2 (PGE2)-targeting cyclodextrins (CDs).

Investigated Object	Investigated Phenomenon	Radiopharmaceutical	Imaging Technique	Reference
healthy BALB/c male mice	diagnostic feasibility, radiochemical synthesis process, proof-of-concept study	[^68^Ga]Ga-NODAGA-HPBCD	in vivo PET/CT imaging, ex vivo radioactivity determination by gamma counter	Hajdu et al., 2019 [31]
BxPC-3 and PancTu-1 tumor-bearing CB17 SCID mice	in vivo and ex vivo organ distribution, PGE2 tumor-homing ability	[^68^Ga]Ga-NODAGA-RAMEB	in vivo static and dynamic PET imaging, ex vivo gamma counter measurements	Trencsényi et al., 2020 [33]
CB17 SCID male mice bearing BxPC-3 tumor	ex vivo biodistribution, PGE2-targeting capability, pharmacokinetic profile, biosynthesis	[^68^Ga]Ga-DOTAGA-RAMEB [^205/206^Bi]Bi-DOTAGA-RAMEB	in vivo static and dynamic PET acquisition, ex vivo gamma counting	Csige et al., 2022 [34]
HT1080, -A20, -PancTu-1, -BxPC-3, -B16-F10 tumor-bearing CB17 SCID male mice and Ne/De, He/De tumor-bearing Fischer-344 female rats	in vivo and ex vivo biodistribution pattern, tumor-targeting competence	[^68^Ga]Ga-NODAGA-RAMEB, [^68^Ga]Ga-NODAGA-HPBCD	in vivo static PET imaging, ex vivo gamma counting	Szabó et al., 2023 [35]

A20: mouse B cell lymphoma; ^205/206^Bi: Bismuth-205/206; B16-F10: mouse melanoma; BxPC-3: human pancreas adenocarcinoma; CD: cyclodextrin; DOTAGA: 1,4,7,10-tetrakis(carboxymethyl)-1,4,7,10-tetraazacyclododecane glutaric acid; PancTu-1: human pancreas adenocarcinoma; ^68^Ga: Gallium-68; He/De: rat hepatocellular carcinoma; HPBCD: 2-hydroxypropyl-β-cyclodextrin; HT1080: human fibrosarcoma; Ne/De: rat mesoblastic nephroma; NODAGA: 1,4,7-triazacyclononane-1-glutaric acid-4,7-diacetic acid; PET/CT: positron emission tomography/computed tomography; PGE2: prostaglandin E2; RAMEB: randomly methylated β-cyclodextrin; SCID: severe combined immunodeficient.

## Data Availability

The datasets used and/or analysed during the current study are available from the corresponding author upon reasonable request.

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
