# Peer review of "Overview of Prostaglandin E2 (PGE2)-Targeting Radiolabelled Imaging Probes from Preclinical Perspective: Lessons Learned and Road Ahead"

_ijms, 2023, doi:10.3390/ijms24086942_

Round 1
Reviewer 1 Report
The manuscript submitted by Kepes Z, et al. is a review article for the development of prostaglandin E2-targeting radiotracers for cancer imaging. Since this is a review article, the topic should be in the general interest of journal readers and supposedly there should be many groups working on the development of prostaglandin E2-targeting radiotracers, and there should be many publications selected for review. However, after reading this manuscript, it is evident that the authors’ group is the only one working in this field, and there are only 4 papers listed in Table 1 from their group cited for review in this manuscript. Therefore, I think it is too premature to select this topic for a review article. In addition, I do not think this manuscript is well written. As this is a review article for radiotracer development, I would expect there will be several figures illustrating the PET (or SPECT) images and chemical structures of the radiotracers. However, there is no figure at all, and there is only one single table listing their 4 papers.
Reviewer 2 Report
please see attached

Reviewer 3 Report
The authors provided a thorough review of the in vivo results of radiolabeled PGE2-directed CDs. While the paper is well-written and organized, it would greatly benefit from the inclusion of chemical schemes, PET imaging, or other related figures. Without these visual aids, readers may find it difficult to follow the research. Therefore, I recommend that the authors add corresponding figures to the manuscript prior to publication.
Reviewer 4 Report
Dear Editor and authors,
Interestingly, the manuscript brings the prostaglandin pathway back into the spotlight. Based on previously established data concerning the prostaglandin pathway and its relationship with cancer, the authors intend to update some new –although still limited– preclinical data for cancer imaging implementation concerning specific prostaglandin-targeting radiolabelled probes. In vitro, in vivo and ex-vivo experimental data are critically evaluated to present the stability, binding affinity, tumour-homing and biodistribution characteristics of the radiolabelled prostaglandin E2-targeting cyclodextrins reported in table 1 in different cell line carcinomas and lymphomas. According to the authors, these biomarkers could be proposed as possible novel diagnostic molecular probes.
In my opinion, the paper is well-written and could be accepted for publication as it is.
Sincerely yours, VL
Round 2
Reviewer 1 Report
This is a revised version and the authors have provided satisfactory changes and responses based on reviewers' comments. Therefore, this revised manuscript could be accepted for publication without further changes.
Reviewer 2 Report
the author has correct the 1st review comments
Reviewer 3 Report
The authors have addressed all the comments. The response looks great. But I do not see any revised manuscript uploaded.